# OpenReview forum: "This State Looks Like That: Self-Interpretable Reinforcement Learning Agents using Prototype Soft Actor-Critic"
_ICML.cc/2026/Conference — ICML 2026 regular_

### Official Review · Reviewer_qjw7 · 2026-03-07

**Soundness:** 2
**Presentation:** 3
**Significance:** 2
**Originality:** 3
**Overall Recommendation:** 4
**Confidence:** 3

**Summary:**

The paper presents ProtoSAC, a method of self-interpretable deep reinforcement learning, which integrates prototype-based reasoning directly into the Soft Actor-Critic (SAC) framework. ProtoSAC generates actions as similarity-weighted combinations over learned prototypes, each associated with a Gaussian action distribution.

**Compliance With Llm Reviewing Policy:**

Affirmed.

**Final Justification:**

The paper makes solid contributions to prototype based interpretability methods, but greatly overstates their explanatory power.

**Key Questions For Authors:**

How did you produce the visualizations in Figures 2, 3 and 8? How can you know what states the prototypes encode after training? Is the encoder invertible? Even if it is, is there a guarantee the outcome is a valid world state? Are those perhaps the initialization states? If so, do the prototypes not shift during training, rendering them obsolete?

How do you arrive the number of prototypes? If prototypes are meant to represent archetypical states, then the right number of prototypes is fundamentally a question about the complexity of the state space partition required to solve the task. It should also connect to RL complexity measures. The prototypes, as presented, make unclear whether they are mere tools to improve performance, or a genuine measure of task complexity.

**Limitations:**

The main limitation of the paper is the strong interpretability claim, which does not follow from its prototype-based method. There is also a question of scalability not addressed by the experiments. If number of prototypes does scale with task complexity, then even finding the right number with hyper parameter tuning (the paper offer no principled alternative, which would be ideal) could be infeasible.

**Strengths And Weaknesses:**

Soundness: The RL methods presented are sound, the loss terms are well motivated. The only issue lies in the strong interpretability claim, which is overstated.  The genuine transparency contribution is limited to the final action being a linear combination of prototype-associated Gaussians, with human-readable weights. However, both the encoder and the similarity computation is opaque, undermining the paper's case for an interpretable method. Prototypes and similarity metrics depend on the embedding space, and the paper never demonstrates that this space has structure meaningful to a human observer, which an interpretability claim would require.

The prototypes are initialized as embedding of real states, which is indeed interpretable. However, both the encoder and the prototype matrix are (I believe) optimized during training, shifting the representation away from their interpretable origin. It is uncertain whether the resulting prototypes even refer to real world states.

Presentation: The paper is generally well-written and clearly structured. It includes a thorough elaboration of the RL context and background methods like SAC. A minor weaknesses is the ambiguity of what the visualizations actually show. Is it initialized states or reverse mapped states of learned prototypes? That is never clarified. The paper also uses terms like "transparent" and "intuitive" loosely and inconsistently throughout, which inflates the interpretability framing beyond what is demonstrated. The method offers some form of decomposability and transparency, rather than interpretability. The figures also do not make a strong point of interpretability.

Significance: The performance results are genuinely significant. Their method matches SAC while outperforming the only comparable self-interpretable baseline across most environments. However, the authors pose the absence of imitation learning distilation from a black box  agent, as unconditional improvement, while one could argue their method produces both a frailer interpretability claim and slower convergence/higher variance.

Originality: To the best of my knowledge this is a novel approach. From the idea of integrating a prototype based actor into SAC, to the specific architecture, the paper presents an original design.

---

> ### Author Rebuttal · Authors · 2026-03-27
>
> - **Concerns related to the visualization and interpretability of the prototypes.**
>
> We thank the reviewer for the comments. As the above questions are highly interlaced, we will provide a combined answer to them. The concerns raised primarily pertain to the nature of prototype-based explanations. Prototype-based methods rely on the assumption that similarity in embedding space reflects similarity in input space. However, we note that this is a general limitation shared by all prototype-based approaches, including PW-Net, Shared-PW-Net, and related work, rather than a specific shortcoming of our method.
> The primary contribution of our work does not aim to advance explainability beyond what prototype-based methods currently offer. Instead, we address a different dimension: the learning paradigm. Unlike prior approaches that rely on imitation learning from a black-box teacher model, our method enables the agent to learn directly from the environment via reinforcement learning. This shift yields performance improvements that are independent of the explainability mechanism.
> Concerning interpretability, our formulation introduces two notable advantages. First, the use of learned distributions parameterized by $\tau$ allows the model to automatically determine the number of prototypes needed, up to a set upper bound. Second, our fidelity analysis reveals that the prototypes learned by ProtoSAC are more decisive: removing the most active prototype at a given state substantially alters the model's decision, confirming that the prototype is meaningfully leveraged rather than incidental. Finally, regarding prototypes' visualization, following the approach of Shared-PW-Net, representative states are identified by selecting the environment frame whose embedding is closest to the corresponding learned prototype.
>
> - **Clarifications regarding the choice of N.**
>
> The initialized number N is an upper bound. The values N=30 (and N=60 for InvertedPendulum) are selected via grid search and represent a safe ceiling: we intentionally initialize more prototypes than we expect to need. The $\tau$ Regularization Loss (Eq. 13) then penalizes the total importance mass across all prototypes throughout training, driving the model to concentrate explanatory weight on the smallest subset sufficient to solve the task. Prototypes with near-zero $\tau$ become functionally inactive and can be disregarded entirely (Fig 4).
> At the same time, a higher upper bound also serves an explicit exploratory function during training. In complex environments, having a larger pool of initialized prototypes allows the agent to cover a broader region of the state space in the early phases of training, before the regularization loss has had sufficient time to prune the inactive ones. This is analogous in spirit to over-parameterization in supervised learning: a larger initial capacity facilitates exploration of the loss landscape, while regularization progressively compresses the representation toward the minimal sufficient structure. For simpler tasks such as Pendulum, a small N suffices from the outset; for higher-dimensional environments such as InvertedPendulum, a larger N=60 is beneficial precisely because the early-training state space coverage problem is harder. For this reason, the effective count is connected to task complexity. Additionally, the sensitivity analysis in Figure 4 directly demonstrates this: for Pendulum (a simple, low-dimensional task), performance is fully retained with as few as ∼5 prototypes; for LunarLander (higher-dimensional, more dynamic), approximately 10 are needed before performance degrades. This ordering is consistent with an informal notion of behavioral complexity.
>
> On connecting to formal RL complexity measures, a rigorous lower bound on the number of required prototypes could, in principle, be derived from covering number arguments over the MDP's state space, or from the number of modes in the optimal Q-function. We acknowledge that establishing such a formal connection is beyond the scope of this paper, but we agree it represents a valuable theoretical direction.
>
> On whether prototypes are tools for performance or measures of complexity: we argue they are genuinely both, and that this is a feature rather than an ambiguity. Prototypes are optimized to solve the task (via $L_\pi$) while simultaneously being regularized to be minimal in number ( $L_\tau$ ), diverse in coverage ( $L_{orth}$  ), and collectively activated rather than degenerate ( $L_{ent}$ ). The emergent effective count is therefore not arbitrary: it reflects the minimum behavioral complexity the policy needs to express.

---

> > ### Author Rebuttal · Reviewer_qjw7 · 2026-04-02
> >
> > I am well aware of the limitations of prototype methods for interpretability, but the acknowledgement given in the rebuttal, is not present in the paper. Instead you make claims like "enabling intrinsic interpretability in continuous action spaces", "By embedding interpretability directly within the policy learning framework, our approach provides intrinsic interpretability from scratch", "self-explainable agents are designed to make their reasoning inherently transparent, offering higher fidelity in the explanations and stronger guarantees of alignment between the model's behavior and the provided interpretations", which are false, as only the aggregation is interpretable.
> >
> > "providing transparent decision-making without sacrificing performance", "Actions are generated as a similarity-weighted mixture over these prototypes, providing transparent and case-based decision-making", yet
> > transparency here means you can see which prototypes contributed and by how much. But you cannot see why a given state is similar to a given prototype, because similarity is computed in latent space. The decision process is decomposable, not transparent. These are meaningfully different things.
> >
> > "our approach employs prototypes for case-based reasoning to enhance interpretability", but genuine case-based reasoning requires that the cases being referenced are meaningful and human-understandable, not embeddings.
> >
> > "which naturally provides intuitive explanations to human users" same problem.
> >
> > "This advancement significantly bridges the gap between high-performance DRL and safety-critical applications, where verifiable and transparent decision-making is essential", the limitations of prototype based methods acknowledged in your rebuttal contradict this statement.
> >
> > "The model computes its output based on the similarity between the input and these prototypes" and immediately following: "which naturally provides intuitive explanations to human users". Here is the main problem. Computing output based on similarity does not naturally provide intuitive explanations unless the similarity itself is interpretable.

---

> > > ### Author Response · Authors · 2026-04-04
> > >
> > > We thank the reviewer for these thoughtful comments and for highlighting an important distinction between decomposable prototype-based decisions and full transparency. We agree that prototype-based methods do not provide full transparency in the strong sense, and that the interpretability they offer is inherently limited by the latent-space representation.
> > >
> > > While we acknowledge the reviewer's concerns, we note that such terminology ("intrinsic interpretability," "transparent decision-making") is well established in prototype-based approaches [a,b,c]. Nevertheless, we completely agree that greater precision is warranted, and we will revise the manuscript accordingly to avoid any overstatement.
> > >
> > > To address this, we will add a clarifying footnote in the introduction after the sentence "By embedding interpretability directly within the policy learning framework, our approach provides intrinsic interpretability from scratch":
> > >
> > > *"Different meanings have been attributed to interpretability in the literature. Here, we refer to 'interpretable' and 'transparent' in the sense common to prototype-based methods—providing a decomposable decision process and prototype-level evidence for inspection, rather than full transparency of the raw input-to-decision mapping."*
> > >
> > > We will also revise the following statements for clarity and precision:
> > >
> > > - "self-explainable agents are designed to make their reasoning inherently transparent..." → "self-explainable agents are designed to make their reasoning more directly inspectable. This removes the need for additional post-hoc procedures, offering higher fidelity in the explanations and stronger guarantees of alignment between the model's behavior and the provided interpretations."
> > >
> > > - "providing transparent decision-making without sacrificing performance" → "providing more inspectable decision-making without sacrificing performance compared to standard SAC."
> > >
> > > - "Actions are generated as a similarity-weighted mixture over these prototypes, providing transparent and case-based decision-making" → "Actions are generated as a similarity-weighted mixture over these prototypes, providing decomposable and prototype-based decision-making. We thank the reviewer for suggesting the term 'decomposable.'"
> > >
> > > - "our approach employs prototypes for case-based reasoning to enhance interpretability" → "our approach employs prototypes for decomposable reasoning to enhance inspectability."
> > >
> > > - "which naturally provides intuitive explanations to human users" → "which provides prototype-level signals that aid inspection of the policy behavior."
> > >
> > > - "This advancement significantly bridges the gap between high-performance DRL and safety-critical applications, where verifiable and transparent decision-making is essential" → "This advancement represents an important step toward bridging high-performance DRL and safety-critical applications, where more inspectable decision-making is valuable."
> > >
> > > We hope these proposed modifications adequately address the reviewer's concerns and better align our presentation with the recommended precision. We thank again for the valuable feedback.
> > >
> > > [a] This Looks Like That: Deep Learning for Interpretable Image Recognition. Chen et al. 2019 (NeurIPS)
> > > [b] Interpretable Image Recognition by Constructing Transparent Embedding Space. Wang et al. 2021 (ICCV)
> > > [c] PIP-Net: Patch-Based Intuitive Prototypes for Interpretable Image Classification. Nauta et al. 2023 (CVPR)

---

### Official Review · Reviewer_teo5 · 2026-03-12

**Soundness:** 3
**Presentation:** 3
**Significance:** 4
**Originality:** 4
**Overall Recommendation:** 5
**Confidence:** 4

**Summary:**

The work studies an important problem of how actors can make self-explainable decisions. The method, ProtoSAC, is developed based on SAC, where they only modify the actor. In particular, the actor is sampled from a Gaussian distribution, where the parameters are determined by a weighted average of the parameters that correspond to a selection of prototypes. The prototypes are learned during training, where the underrepresented prototypes are replaced every few episodes. They conduct experiments on eight environments, including discrete and continuous control problems. They demonstrate ProtoSAC to enjoy competitive performance while being self-explanatory.

**Compliance With Llm Reviewing Policy:**

Affirmed.

**Key Questions For Authors:**

- I might have missed something. However, in Figure 3, why do you have two distributions? I thought each prototype only corresponds to one distribution. Also, why is (a) empty?
- You only have Figures and Tables for a few environments. Can you have them for all environments?
- I thought you would replace an old prototype with a new one directly through Eq. (12). I don't understand what Eq. (14) is for.

**Limitations:**

yes

**Strengths And Weaknesses:**

## Strengths
- Developing RL methods, where the decisions of the policy are explainable, is important. And to the best of my knowledge, ProtoSAC is the first to be self-interpretable without relying on imitation learning, and is able to learn the prototypes without manual designs. This suggests the novelty.
- The design of ProtoSAC is logically sound, and the conducted experiments support its goals and claims. In particular, Table 2 suggests a comparable performance to its basis, SAC, and the most relevant comparator, Shared-PW-Net. Figures 2 and 3 show the top four learned prototypes and the corresponding action distributions in different tasks. Table 3 does indicate that the most relevant prototypes are indeed representative.

## Weaknesses
- The Pseudocode in Algorithm 1 seems to be only half-done. I.e., you mention how to select actions, but you did not say how to / what to update.
- The paper is mostly well-written and easy to follow. However, there are a few questions that need clarification. Please see the questions below.

---

> ### Author Rebuttal · Authors · 2026-03-27
>
> - **Algorithm 1**
>
> We thank the reviewer for pointing this out. The pseudocode in Algorithm 1 focuses on the action selection mechanism. We will replace the "until convergence" in the algorithm in the appendix detailing the prototype update procedure as follows:
>
>
> ***
>
> **Require:** Environment ${E}$, number of prototypes $N$, update interval $M$, quantile threshold $q$, loss coefficients $\alpha$, $\gamma$
>
> 1. Initialize replay buffer ${D}$
> 2. Initialize critic networks $Q_{\phi_1}$, $Q_{\phi_2}$ and target networks $Q_{\bar\phi_1}$, $Q_{\bar\phi_2}$
> 3. Initialize encoder $f_s$, prototype matrix $P \in \mathbb{R}^{N \times H}$, importance weights $\tau \in \mathbb{R}^N$, means $\mu \in \mathbb{R}^{N \times O}$, standard deviations $\sigma \in \mathbb{R}^{N \times O}$
> 4. **for** each episode **do**
> 5. &nbsp;&nbsp;&nbsp;&nbsp;**for** each environment step **do**
> 6. &nbsp;&nbsp;&nbsp;&nbsp;&nbsp;&nbsp;&nbsp;&nbsp;Encode state: $z_s = f_s(s)$
> 7. &nbsp;&nbsp;&nbsp;&nbsp;&nbsp;&nbsp;&nbsp;&nbsp;Compute similarities $\text{sim}(z_s, p_i)$ via Eq. (8)
> 8. &nbsp;&nbsp;&nbsp;&nbsp;&nbsp;&nbsp;&nbsp;&nbsp;Compute $P_\text{sim}$ via Eq. (10)
> 9. &nbsp;&nbsp;&nbsp;&nbsp;&nbsp;&nbsp;&nbsp;&nbsp;Compute $\mu_{z_s}$, $\sigma_{z_s}$ via Eq. (11)
> 10. &nbsp;&nbsp;&nbsp;&nbsp;&nbsp;&nbsp;&nbsp;&nbsp;Sample action $a \sim {N}(\mu_{z_s}, \sigma_{z_s})$
> 11. &nbsp;&nbsp;&nbsp;&nbsp;&nbsp;&nbsp;&nbsp;&nbsp;Execute $a$, observe $r$, $s'$; store $(s, a, r, s')$ in ${D}$
> 12. &nbsp;&nbsp;&nbsp;&nbsp;**end for**
> 13. &nbsp;&nbsp;&nbsp;&nbsp;**for** each gradient step **do**
> 14. &nbsp;&nbsp;&nbsp;&nbsp;&nbsp;&nbsp;&nbsp;&nbsp;Sample minibatch $(s, a, r, s') \sim {D}$
> 15. &nbsp;&nbsp;&nbsp;&nbsp;&nbsp;&nbsp;&nbsp;&nbsp;Update critics: minimize ${L}_Q(\phi_i)$ via Eq. (5), $i = 1, 2$
> 16. &nbsp;&nbsp;&nbsp;&nbsp;&nbsp;&nbsp;&nbsp;&nbsp;Update actor: minimize full loss via Eq. (16)
> 17. &nbsp;&nbsp;&nbsp;&nbsp;&nbsp;&nbsp;&nbsp;&nbsp;Update target networks: $\bar\phi_i \leftarrow \rho\,\bar\phi_i + (1-\rho)\,\phi_i$
> 18. &nbsp;&nbsp;&nbsp;&nbsp;**end for**
> 19. &nbsp;&nbsp;&nbsp;&nbsp;**if** episode $\mod M = 0$ **then**
> 20. &nbsp;&nbsp;&nbsp;&nbsp;&nbsp;&nbsp;&nbsp;&nbsp;Identify underutilized prototypes: ${U} = \{i : \tau_i < \text{quantile}(\tau, q)\}$
> 21. &nbsp;&nbsp;&nbsp;&nbsp;&nbsp;&nbsp;&nbsp;&nbsp;**for** each $i \in {U}$ **do**
> 22. &nbsp;&nbsp;&nbsp;&nbsp;&nbsp;&nbsp;&nbsp;&nbsp;&nbsp;&nbsp;&nbsp;&nbsp;Select replacement candidate via Eq. (12): $p_i \leftarrow \arg\max_{z_{s_j}} \frac{1}{\sqrt{\sum_{n} \text{sim}(z_{s_j}, p_n) + \epsilon}}$
> 23. &nbsp;&nbsp;&nbsp;&nbsp;&nbsp;&nbsp;&nbsp;&nbsp;&nbsp;&nbsp;&nbsp;&nbsp;Initialize $(\mu_i, \sigma_i)$ from current policy via Eq. (10)
> 24. &nbsp;&nbsp;&nbsp;&nbsp;&nbsp;&nbsp;&nbsp;&nbsp;&nbsp;&nbsp;&nbsp;&nbsp;Reinitialize $\tau_i$ randomly
> 25. &nbsp;&nbsp;&nbsp;&nbsp;&nbsp;&nbsp;&nbsp;&nbsp;**end for**
> 26. &nbsp;&nbsp;&nbsp;&nbsp;**end if**
> 27. **end for**
> ***
>
> - **Fig 3**
>
> Each prototype represents a combination of a state and an action. In the Lunar Lander environment, the action space is composed of two components: the main engine and the lateral boosters throttle, which is why two distributions appear in Figure 3. We apologize for the ambiguous caption and will revise it to avoid confusion.
>
> Regarding subplot (a), it is empty because it corresponds to the landing prototype, where the optimal behavior is to shut down all engines and remain stationary.
>
>
> - **Figures of prototypes for all the environments.**
>
> We thank the reviewer for the suggestion. We will include figures and tables for all environments in the appendix.
>
> - **I thought you would replace an old prototype with a new one directly through Eq. (12). I don't understand what Eq. (14) is for.**
>
> Equation (12) ensures that the prototype candidates are sufficiently different from the current prototype set. However, it does not guarantee diversity among the new candidates themselves. Equation (14) further enforces dissimilarity across all prototypes, including the newly selected ones, ensuring they cover distinct directions in the latent space.

---

> > ### Author Rebuttal · Reviewer_teo5 · 2026-04-02
> >
> > Thanks for the clarification. And after reading also other reviews and replies, I have decided to keep the score.

---

### Official Review · Reviewer_vraY · 2026-03-13

**Soundness:** 3
**Presentation:** 2
**Significance:** 3
**Originality:** 3
**Overall Recommendation:** 4
**Confidence:** 4

**Summary:**

The authors introduce a new model in the realm of explainable AI, here particuarly in deep RL models which are interpretable via prototypes. Other XAI deep RL models using prototypes require using imitation learning, which is kind of supervised. One recent attempt requires the humans to choose the prototypes, while the other doesn't require humans to choose prototypes, but still uses imitation learning. This is the first paper doing XAI deep RL with prototypes with full RL, done via a soft actor-critic model. This model is as performant as the other XAI deep RL models with signals of choosing better, more explainable prototypes.

**Compliance With Llm Reviewing Policy:**

Affirmed.

**Final Justification:**

As noted in my rebuttal comment, no update to my score needed from a 4.

**Key Questions For Authors:**

- Why didn't other authors try soft actor-critic?
- How does SAC actually differ from imitation learning-based approaches in the 2023 related works?
- Could this work for other modalities of RL?

**Limitations:**

Yes

**Strengths And Weaknesses:**

Strengths:
- Development of a de novo model trained via RL instead of imitation learning seems to be a novel advance
- Model is as performant across several different benchmarks and RL settings, which showcases the impressive breadth of this method
- Method of prototype selection, pruning, and generation is well explained and seems to contribute to the powerful result that it improves on Shared-PW-Net in terms of explanability

Weaknesses:
- No discussion of whether ProtoSAC can handle discrete action spaces, whereas Borzillo et al (2023) and Kenny et al (2023) discuss both continuous and descrete action spaces
- Sections 1 and 2 are repetitive --- discussion of prototype-based methods for DRL seems repeated in both sections, as are the contributions of Borzillo et al (2023) and Kenny et al (2023).
- The prototype selection mechanism seems to be influenced by ideas from entropic regularization, and could be fleshed out with a fuller discussion of properties of the performance of this selector

---

> ### Author Rebuttal · Authors · 2026-03-27
>
> - **Discrete action-spaces**
>
> While SAC was originally designed for continuous action spaces, discrete variants have been proposed in the literature (e.g., Christodoulou, 2019), meaning ProtoSAC could in principle be extended to discrete settings. However, the present work deliberately focuses on continuous action spaces, and we leave the adaptation to discrete environments as future work. To better highlight this matter, we will include this as a limitation and future work perspective in the conclusions of the revised manuscript.
>
> [a] Christodoulou, Petros. "Soft actor-critic for discrete action settings." arXiv preprint arXiv:1910.07207 (2019).
>
> - **Sections 1 and 2 are repetitive**
>
> We understand that some concepts might appear repeated. However, we argue that Section 1 (Introduction) and Section 2 (Related Work) serve fundamentally different roles. The Introduction briefly mentions Borzillo et al. (2023) and Kenny et al. (2023) to motivate the research problem and highlight the gap our work addresses, without going into technical detail. Section 2 then revisits these works in the context of a structured and comprehensive review of the literature, providing a deeper methodological analysis necessary to precisely position our contribution.
>
> - **Entropic Regularization**
>
> The reviewer's intuition is correct. The selection mechanism in Eq. (10) is a temperature-scaled softmax where $\tau_i$ acts as a learned inverse temperature per prototype. Two loss terms govern its entropic behavior: $L_\tau$ enforces global sparsity by penalizing total importance mass, while $L_{ent}$ prevents local collapse onto a single prototype. Together, they implement a sparse entropic regularization (few prototypes are used globally, but each state activates a meaningful mixture locally), closely related in spirit to entropic optimal transport and sparse attention mechanisms.
>
> The empirical contribution of each term is already validated in our ablation study (Appendix, Table 4), which shows that removing $L_{ent}$ leads to degenerate hard assignments, while removing $L_\tau$ prevents the model from concentrating weight on a minimal prototype subset.
>
>
> - **Why didn't other authors try SAC? How does SAC actually differ from imitation learning-based approaches in the 2023 related works?**
>
> PW-Net and Shared-PW-Net operate by first training a black-box RL agent to convergence, then distilling its behavior into a prototype network via imitation learning. This transforms RL into a supervised problem where the prototype layer mimics the output of a finished policy: there is no RL loop left to integrate SAC into. The choice of teacher algorithm is entirely irrelevant to their prototype mechanism.
>
> ProtoSAC dissolves this boundary entirely: the prototype-based actor replaces the standard Gaussian policy inside the SAC training loop, with prototypes learned jointly from reward signals and no black-box teacher required.
>
> - **Other RL Modalities**
>
> ProtoSAC, as presented, operates on vector state representations. Extensions to visual RL (with a convolutional encoder for fs), language-conditioned RL, or multi-modal settings are, in principle, compatible with the architecture. This is a promising future direction that we will briefly mention in the Conclusions.

---

> > ### Author Rebuttal · Reviewer_vraY · 2026-04-02
> >
> > My concerns are resolved. I think that to improve my score from a 4 to a 5 I would have to see the authors implement the discrete variant of SAC as discussed in the rebuttal. I believe however that would be outside of the scope of a revision and so will leave my score where it is.

---

> > > ### Author Response · Authors · 2026-04-04
> > >
> > > We thank the reviewer and we understand their position. In this paper our aim was to transform SAC using a prototype-based approach. SAC is primarily for continuous action spaces, and discrete-action space adaptations represent variations of the algorithm. Other approaches are more suited for discrete-action spaces. For this reason, while we stand by the point that future work can adapt ProtoSAC for discrete action spaces, we see it as more valuable to adapt other (more suited) algorithms rather than SAC for interpretable agents on discrete-action spaces. However, we understand the reviewer's point and still thank them for the valuable feedback.

---

### Official Review · Reviewer_s8Ph · 2026-03-20

**Soundness:** 2
**Presentation:** 3
**Significance:** 3
**Originality:** 3
**Overall Recommendation:** 4
**Confidence:** 3

**Summary:**

The paper presents an approach for more interpretable RL policies by combining "prototypes" describing pairs of (state, action distribution). In particular, the idea is to identify similar-looking states within the prototype set, then statistically combine the action distributions.

**Compliance With Llm Reviewing Policy:**

Affirmed.

**Final Justification:**

This paper seems to propose a novel method, which both appears to be potentially explainable (as advertised) and performant enough to be competitive with alternatives. While the results are uneven in terms of exceeding the performance of the alternatives, I don't think that's a huge problem, and perhaps offer valuable opportunities for follow-up work.

**Key Questions For Authors:**

1. In related work "However, a significant limitation... necessity of pre-labeled datasets...". Is this really true? Bottom-up clustering is a common way to avoid the need for datasets, at the cost of creating a different problem: attributing meaning to clusters of states.
2. "Experiment setup" indicates that the program used 30 prototypes for most environments and 60 for another, but does not provide intuition on how the reader should choose these numbers at the outset (i.e., Figure 4 seems to be doing a search on the range 1-N, where here N=30, usually; it does not give a decision calculus for choosing the 30 initially). Similarly, given that the agent started with so many prototypes (and Figure 4 indicates mountain car needs about 10), the figures feel a little insufficient showing only 4.
3. (follow on to previous question) For Figure 3, the prototypes look different from what I'd expect. In particular, #1 and #2 are both near the ground by the reader's left-hand flag, but there is no prototype with the lander far from the ground near that same flag. Why is this?
4. I'd be willing to potentially raise my score if the rebuttal can prepare the necessary statistical analysis and rework the necessary reporting about those results.

**Limitations:**

The paper does not include the boilerplate impact statement, instead describing the contributions' value, meaning the paper has missed the mark for the conference's intent for that paper section.

**Strengths And Weaknesses:**

Strength - Approach seems to work from a performance standpoint

Strength - Paper is well-written

Weakness - Paper makes statistical claims without statistical evidence

Specific Issues
---------------

- Paper is well-written

Introduction "While post-hoc methods... they do not reflect the true internal decision-making process of the model but instead approximate it." This is a critical point, one that many researchers underappreciate

- Paper makes statistical claims without statistical evidence

S5 starts off indicating comparative statistics are coming. Later, we see this "...ABC outperforms XYZ by a significant margin..." The next 10 lines make a number of statistical claims of this nature, in formal language as quoted. The problem is that there aren't any formal statistical tests with p-values, effect sizes, and such to back up these formal claims! (sidenote: there are more of these in Appendix D "differences are marginal"). I also found the dismissal of negative results (e.g., HalfCheetah) to be more flippant than is appropriate. This is an inroad to discuss where and how people should deploy this algorithm. My sense from reading the paper is that the answer to this question is that this algorithm will do well when the state representation is highly compressible, in terms of the truthfulness of the statement that the agent should behave similarly in states that appear similar.


Minor Issues
------------

- All of the XDRL section of related work is one paragraph, that spans about 50 lines. There is another massive paragraph later for "experiment setup"

- Paper is inconsistent about including sentence punctuation in Eqs. Eq 1 does not have sentence punctuation, while 2, 3, and others do.

- There is an O after Eq 8 that I think should have math mode.

- Velocity lines ("Red sparks" indicated in the prose) are not visible enough in the figures. Consider amplifying that in some way.

- References are untidy. Check for proper nouns and acronyms (e.g., RL)

- There is a weird hyphenation issue in figure 8's caption

---

> ### Author Rebuttal · Authors · 2026-03-27
>
> - **Unsupported Claim:** We thank the reviewer for pointing this out. We acknowledge that this claim is missing support. We propose to rephrase it as follows: "While post-hoc methods can offer insights into agent behavior, they are generally less faithful (Rudin, 2019), as most of them (Bastani et al., 2018; Verma et al., 2018; Frosst \& Hinton, 2018), do not reflect the true internal decision-making process of the model but instead approximate it."
>
> [a] Rudin, C. "Stop explaining black box machine learning models for high stakes decisions and use interpretable models instead." Nat. Mach. Intell. (2019).
>
> - **Statistical significance:** We thank the reviewer for this observation. We have performed Welch's t-tests and computed Cohen's *d* effect sizes for all pairwise method comparisons, using the 30 simulation runs already reported in Table 2. The full results are provided below.
>
> | Environment | Comparison                | Mean Diff | t-stat | p-value    | Cohen's d |
> | ----------- | ------------------------- | --------- | ------ | ---------- | --------- |
> | Pendulum    | ProtoSAC vs SAC           | −10.45    | −0.55  | 0.584 (ns) | −0.14     |
> | Pendulum    | ProtoSAC vs S-PW-Net | +372.09   | 5.15   | <0.001 *** | 1.33      |
> | LunarLander | ProtoSAC vs SAC           | +25.83    | 1.56   | 0.128 (ns) | 0.40      |
> | LunarLander | ProtoSAC vs S-PW-Net | +23.65    | 2.21   | 0.034 *    | 0.57      |
> | MountainCar | ProtoSAC vs SAC           | −2.26     | −15.17 | <0.001 *** | −3.92     |
> | MountainCar | ProtoSAC vs S-PW-Net | −2.34     | −15.51 | <0.001 *** | −4.01     |
> | InvPendulum | ProtoSAC vs SAC           | 0.00      | —      | —          | 0.00      |
> | Hopper      | ProtoSAC vs SAC           | +5.97     | 3.60   | 0.001 **   | 0.93      |
> | Hopper      | ProtoSAC vs S-PW-Net | +11.81    | 6.91   | <0.001 *** | 1.78      |
> | HalfCheetah | ProtoSAC vs SAC           | −1223.15  | −39.62 | <0.001 *** | −10.23    |
> | HalfCheetah | ProtoSAC vs S-PW-Net | −493.99   | −2.16  | 0.039 *    | −0.56     |
> | Humanoid    | ProtoSAC vs SAC           | +379.75   | 1.53   | 0.137 (ns) | 0.40      |
> | Humanoid    | ProtoSAC vs S-PW-Net | +4457.23  | 183.46 | <0.001 *** | 47.37     |
> | CarRacing   | ProtoSAC vs SAC           | +136.08   | 3.14   | 0.003 **   | 0.81      |
> | CarRacing   | ProtoSAC vs S-PW-Net | +401.58   | 11.60  | <0.001 *** | 3.00      |
>
> *\*p<0.05, \*\*p<0.01, \*\*\*p<0.001. InvertedPendulum: both SAC and ProtoSAC achieve a perfect score of 1000±0.0 in all 30 runs; a t-test is undefined.*
>
> We will incorporate these results in the appendix and improve the comments in the revised manuscript.
>
> - **Dense Paragraphs and Images (sparks):** We acknowledge that the paragraphs are dense. Thanks to the additional page in the revised manuscript, we plan to add extra spacing in dense paragraphs and to enlarge the images. Also, we will include full-size images as supplementary material.
>
> - **Typos:** We thank the reviewer for pointing out these problems. Indeed, Eq. 1 is lacking a full stop, the O after Eq. 8 should be in math mode, and there is a typo in the caption of Fig 8. We will fix these typos and make sure that references and acronyms are tidy in the revised version.
>
> - **Pre-labeled concepts and clustering:** Concept-based methods need pre-labeled datasets as they do not use clustering but rather the identification of human-interpretable concepts. Clustering is more of a prototype-based approach, as it relies on specific instances (prototypes), and adding human-interpretable labels would still require manual attribution.
>
> - **Prototypes:** The number of prototypes is selected via grid search over a predefined range. We will specify this detail in the revised manuscript. Additionally, we invite the reviewer to refer to the reply to reviewer qjw7 for further details on such matters. Regarding Fig 4, due to space constraints in the main paper, we selected the most representative and informative prototypes for illustration. We will add the complete set of figures for all environments in the appendix.
>
> - **Prototype 1 in Fig 3:** P1 represents a state where the ship is about to stop, and there is no need for the model to "act". This is interesting as if the model had only learnt cases in which engines should be on, it would never know how to stop.  On the other hand, in P2, the ship's speed is high, and it must counterbalance it to avoid collisions. Moreover, Fig 3 shows only the top-4 prototypes to offer an idea of the model's behavior. As correctly noted, to have a complete picture, one should inspect all the prototypes. Indeed, in Fig 4, we show that for LunarLander, the top-10 prototypes are needed to solve the task. For this reason, Fig 3 might lack some expected behaviors. Space constraints do not allow us to show all the prototypes in the main paper. To improve the model inspection, we will include the other prototypes in the appendix of the revised manuscript.

---

> > ### Author Rebuttal · Reviewer_s8Ph · 2026-04-02
> >
> > Stats workup is a good start, I have a little more feedback on that. But I think it is possible to accept the paper in its current form, so I'll increase my score by 1 to a weak accept.
> >
> > Additional thoughts after reading the rebuttal:
> > 1. Be careful not to violate assumptions of tests. Welch's is a good choice in that it does not assume equivariance, but it does still assume normality.
> > 2. I'd organize the table differently; the goal is to compare against the algorithms, so sort on that.
> > 3. There are a lot of tests here, meaning it might be worthwhile to also show both p-values and effect sizes in raw form in addition to after some statistical correction.
> > 4. There are a lot of large effect sizes here, some of which point in different directions. Please use this in revising the discussion to unpack why the results might be inconsistent across domains.
> > 5. I still find the "grid search across a pre-specified range" to be insufficiently motivated. While I'm not looking for strong theory on making a principled choice, procedures are also good. For example, suppose a user chose a range that was too small. How might they discover that? If the answer is something general like "performance degrades", how can they isolate the cause as being insufficient prototypes, as opposed to some other reason performance could degrade?

---

> > > ### Author Response · Authors · 2026-04-05
> > >
> > > We thank the reviewer for the continued constructive feedback. We will carefully follow all  recommendations in the revision.
> > >
> > > Regarding prototype number selection via grid search over a pre-specified range, we acknowledge this could be better motivated. However, this follows standard practice in prototype-based methods (e.g., ProtoPNet selects N via validation performance on a held-out set). The choice of hyperparameters is highly dependent on the environment, and more sophisticated techniques for automated selection (e.g., meta-learning) represent a promising direction for future work.

---

### Decision · Program_Chairs · 2026-04-30

**Decision:**

Accept (regular)

**Comment:**

Generally, reviewers found this paper reasonably technically sound, reasonably well-written, and novel. It may be useful to at least some fraction of the ICML community. There were some concerns about statistical significance (part of the claimed contributions) that were partially addressed.